# To Make Known in Order to Recognize: Schools as Vehicles for Constructing Identity for the Maya Peoples

Monia Rodorigo *, Javier González-Martín * and Susana Fernández-Larragueta

Department of Education, University of Almería, 04120 La Cañada de San Urbano, Almería, Spain; sfernan@ual.es
* Correspondence: mrodorigo@ual.es (M.R.); jgonzal@ual.es (J.G.-M.)

**Abstract:** Although the Guatemalan population is made up of 53% Indigenous peoples, there is a certain belief according to which Indigenous peoples are still currently associated with underdevelopment, inferiority, submission, and exploitation, causing the belief that, for many, being Mayan is not synonymous with identity, but with poverty, neglect, and inferiority. Through a biographical–naturalist study, we delve into how Guatemalan teachers experienced not only their Mayan being, but also how the construction of their identity was gestated in schools. After carrying out the data analysis process, we underline and describe how the Mayan culture, worldview, and language still constitutes a difficulty and an obstacle to the construction of the personal identity of Indigenous peoples in Guatemala. We describe the experience lived in monolingual and monoethnic schools in which not only is the Mayan culture not recognized, but the use of the language and its expression are prohibited. We review the repercussions that the actions of the political agendas have on the key of recognition, highlighting how, beyond the legislative indications, it is the involvement and personal actions of teachers that is making it possible to make known and recognize the Mayan culture.

**Keywords:** cultural identity; intercultural education; inclusion; educational policy; Mayan culture; primary and secondary school



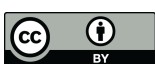

## 1. Introduction

Until the 1990s, the term Maya was exclusively used by historians, anthropologists, archeologists, and linguists who studied the customs and habits of native Indigenous peoples, mainly in the territories that are now known as Mexico, Guatemala, Belize, Honduras, and El Salvador. However, it has evolved into a movement that implies a new way of recognizing and understanding the political participation of the "indigenous" or, more specifically, the "Maya peoples". A term that, after almost three decades of vindictive use, is now defined as a "culturally differentiated collective with its own history, habitually used in certain political and academic spaces, being nowadays the politically correct way of mentioning this population" [1] (p. 197). The Maya peoples cannot be understood without perceiving how the label "Maya" has been identified, explained, and established throughout history as a product of historical constructions generated as a product of contact between groups of different origins [2] with different levels of power, and used by some to justify their dominance over others.

It should be noted that Guatemala is a multiethnic, multicultural, and multilingual state, where 43.6% of the population identifies as Indigenous or Afro-descendant peoples. Among them, 41.7% are Mayan, 1.8% are Xinka, and 0.1% are Garifuna [3]. Article 143 of the Political Constitution of the Republic of Guatemala (1985) considers Spanish as the official language, while declaring the need to legally recognize all the national languages that coexist in order to facilitate the conservation and development of their own cultures for the common good and to improve the quality of life of all inhabitants of the country (Law of National Languages, 2003). This approach strives to legitimize "ethnicity" as a tool for liberation, countering the prevailing inequality within a social environment where

"white" and "Western" are often seen as teleological endpoints [4], contrasting with the notion of "traditional" ethnicity [5].

Our research focus on Guatemala is based on its linguistic reality in which, in addition to Spanish, 22 Mayan, Xinka, and Garífuna languages are spoken. These are languages that must be respected, promoted, improved, and used according to the particularities of each linguistic community. Historically, ethnic minorities have been socially marginalized due to unequal territorial and socioeconomic conditions in the region. Indeed, as was already mentioned, low living standards and social discrimination have historically made them victims of rejection and denial. Many of these groups live in rural areas that are far from the main educational centers, with scarce and inadequate local offerings in terms of infrastructure, maintenance, quality of teachers, and teaching materials [6].

In this context, it is crucial for the Guatemalan administration to respond effectively to the diverse realities of a country with such a high degree of diversity. The government must fulfill its obligations, as outlined in the Peace Agreements (1991–1996), to ensure significant progress is made in matters related to the defense of languages and the promotion of intercultural bilingual schools, as well as authentic and meaningful representation of the "Mayan people" in all public institutions. However, the reality is more complex and intricate than a few consensual documents, so it is necessary to reflect on the construction of the identity of the "Maya peoples" so that they can undergo a genuine process of social recognition.

## 2. The Cultural Identities of the Maya Peoples and Their Construction at School

Romero Rodríguez et al. [7] found that the cultural identities of different groups are not solely constructed based on a single variable, but rather through the interaction of a set of dimensions, such as age, gender, religion, social class, ethnicity, etc. Moreover, these dimensions do not always interact in the same way, and it is impossible to select one of them as the most relevant. However, their analysis must be based on how they qualify and redefine life experiences [8].

Before diving into the "childhoods of the Maya people", it should be said that cultural identities begin to develop in childhood through various life experiences, with the school being a singular place for this. The interaction between students, teachers, and social experiences nurtures this process, which is nuanced by linguistic, religious, or physical differences that gradually shape cultural representations and stereotypes [9]. These elements progressively intertwine throughout childhood as if they were chapters in a book, which together form the construction of a dynamic cultural identity in which emotions and lived events intermingle, giving shape to what Jovchelovith defined as cultural heritage [10].

When referring to intercultural childhoods such as those of Guatemala, which is multiple, varied, and multilingual within the same country, it is crucial to analyze how cultural identities develop when national and patrimonial cultures are confronted and when Indigenous groups, upon their arrival at school, are devalued and blurred by educational institutions. Therefore, the way in which they self-determine as Indigenous is controlled and redefined through state-promoted educational policies, building bureaucratized and symbolic ethnicities that ignore the importance of languages, cultural principles, and ancestral knowledge [11]. If identity is a dialogue between norms and contraries [12], children who are aged 5 or 6 years begin to know the degree of acceptance among peers and with adults, while internalizing their first social and relational norms among which they would find identity, including race [13].

Although education is a right that is enshrined in the Political Constitution of the Republic of Guatemala (1985), Article 33 of the National Education Law stipulates that the state must provide education to its citizens without discrimination (National Education Law, Decree 12–91). Likewise, according to Article 76, educational centers located in areas with predominant Indigenous populations should prioritize bilingual education to ensure that the right to education is fully exercised and not conditioned by an individual's social origin, territory, ethnicity, or gender. Although these transculturation processes

are common and standardized in urban schools, this is not the case in rural areas, which instead base their strategies on what can be defined as a monocultural curriculum that does not take into account the multicultural reality of the country [14]. This curriculum does not incorporate the Mayan culture and worldview, nor does it provide access for those whose native language is a Mayan language rather than Spanish.

As mentioned above, this reality is based on power relations that hinder intercultural bilingual educational environments in addition to depending on the country's socioeconomic inequality. This transforms the school into a space that reproduces socioeconomic and cultural inequalities and asymmetries among students [15]. They promote socio-cultural models and pedagogical practices that "homogenize identities and propose for transformation and abandonment of the native culture" [14] (p. 87).

In terms of school enrollment, 71.9% of Guatemalan students are enrolled in public schools, 23.4% are enrolled in private schools, 4.1% are enrolled in cooperatives, and only 0.5% are enrolled in municipally owned schools. Access to pre-primary education is often delayed until the age of five or six. However, by the age of seven, school enrollment reaches 100% [16]. The delay in school enrollment primarily stems from the high percentage (17.9%) of children whose mother tongue differs from the language offered by the school. This delay in schooling is later compounded by an accelerated dropout rate between the ages of 13 and 18, with the rates of children remaining in education being relatively low at only 54.1%, 43.7%, and 26.5% for 16-, 17-, and 18-year-olds, respectively. In fact, although the overall dropout rate is 51%, if we only analyze the variable of the Indigenous population, this rate increases to 65%.

We would also like to point out that, if these numbers already seem worrying, when analyzed at a territorial level, the situation becomes even more alarming when we focus on rural areas with a majority Indigenous population. Although an intercultural bilingual education model that "establishes a theoretical-conceptual approach to basic curricular content and a linguistic policy that served as the basis for the development of texts in Mayan, Garifuna, and Xinka languages" [17] (p. 19) has existed in Guatemala since the year 2000, there is a lack of a network of nearby public schools that offer free primary and secondary education that responds to these communities' social, linguistic, cultural, and economic needs. This represents a significant barrier to the retention and access to higher education. Despite aiming to promote coexistence and providing tools for the four Indigenous peoples (Ladino, Mayan, Garifuna, and Xinka) to enhance their opportunities for local, regional, and national growth and to fully develop their potential in various social domains for genuine intercultural coexistence, the current system falls short. Although the system has reduced the school dropout rate by increasing the number of bilingual teachers in municipalities with a significant Indigenous population, it is only applied up to the third grade in primary school and only in certain areas of the country, diminishing and inhibiting its main potential and making it clear that there is a need for a deeper analysis of it.

## 3. Materials and Methods

The research presented here is part of the International Cooperation Project "Teachers for Mayan girls and boys in remote rural areas: Training the Trainers" (2018UC006) funded by the Agencia Andaluza de Cooperación Internacional para el Desarrollo de la Junta de Andalucía-Spain. The project was conducted in Alto Verapaz, Guatemala, and carried out between the University of Almería (UAL) and the Instituto Técnico Maya de Estudios Superiores (ITMES). The project had the following two main objectives: firstly, to train future university teachers to work with highly vulnerable students from rural Mayan areas, and secondly, to establish a curricular framework that integrates intercultural teacher training with the incorporation of Mayan vision and gender perspectives.

Therefore, the main purpose of the presented study is to determine the fundamental aspects that teachers identify about Mayan cosmovision and how they, through their school tradition, perceive that their Mayan cultural identity is treated in their school context. The teachers involved were aware that cultural identities are constructed from a more or less

numerous set of elements and that school "is a catalyst in the construction of such cultural identity(ies) to the extent that it is negotiated between students and teachers" [7] (p. 484). Consequently, this study focuses on the following three aspects: (a) to know the perception of participants about being and forming part of the Maya peoples as an element of identity construction; (b) to describe their perceptions about the influence that the school has had on their identity development (personal and professional); and (c) to dive into the explicit and implicit meanings of educational policies for rural Mayan communities.

Using the biographical method, we approached the deepening and analysis from a naturalistic and interpretative paradigm [18], allowing us to enter into the participants' constructs about the formulation of their world [19–22]. To this end, 16 Mayan teachers were selected in a prospective sampling process. The main criteria were (a) that they were leaders of their communities and (b) that they were teachers in Mayan intercultural contexts in both rural and urban areas. We understood that "to analyze those problems or situations that present multiple variables and that are closely linked to the context in which they develop" [23] (p. 667), it is essential to study the concrete educational realities in which they occur. These teachers were selected so that we could learn first-hand about their experiences as intercultural teachers and referents in their communities, committed to spreading the Mayan culture. We also sought to learn about and reflect on how schools are currently supporting or inhibiting the identify development of a group as large as the Indigenous Mayan population in Guatemala in the face of rejection caused by a hostile environment, and how schools could support or inhibit identity development in the future. Similarly, as Mayans, we were interested in delving into their educational biographies, reflecting on how they have experienced their own journeys as students in educational institutions.

The information was gathered in two phases over a six-month interval, with the necessary participation of the ITMES team, on the one hand, due to the difficulty of accessing the terrain and the languages—Q'eqchi' and Poqomchi'—and, on the other hand, the UAL researchers in the field in Guatemala, due to their experience in developing naturalistic and interpretative research and the stated methodology. The instrument used was mainly semi-structured and in-depth interviews with selected participants, representatives of the rural communities visited, and informal conversations with other teachers and those responsible for the institutions visited.

In the first phase, we conducted interviews with five female teachers and five male teachers, while in the second phase, we returned to the field to re-interview three of the initial participants and collected information from two more female teachers and four male teachers. For the analysis of the information, we followed a process of extracting emerging themes grouped into broad categories [24] and, following a discursive analysis, we first created singular narratives and then cross-referenced narratives that are presented in this text.

To ensure the transferability of the qualitative research [18], the discussion of the data incorporates relevant information that is provided directly by them and is presented in quotation marks with reference to the person who was interviewed.

## 4. Results

The results of our research and the categories extracted from the data analysis process that was carried out during the research are described below, first in a table (Table 1), where we explain their relationship and construction more clearly, and then in points 3.1, 3.2, and 3.3 through a theoretical and practical discussion.

**Table 1.** Main theme, categories, and subcategories.

| Theme | Major Categories | Minor Categories |
|---|---|---|
| Cultural issue | Mayan culture | Cosmovision<br>Being Mayan as a feeling<br>Mayan symbols as elements of identity<br>Practice rituals |
| | Languages | Language as a symbol of identity<br>The language of the school<br>Learning Spanish as a space for social mobility<br>The complexity of languages |
| School environment | Background | School does not recognize Mayan culture<br>Emotional conflicts at home/school<br>Lack of specific content in CVs<br>Risk of losing identity<br>Lack of commitment from teachers in rural areas |
| | People's role | Teachers who help to recognize identity<br>The lack of cultural presence as a push for vindication<br>People as a tool to publicize and recognize the Mayan culture<br>Lack of individual responsibility |
| | Administration issues | School as a privileged space to transfer the Mayan culture<br>Mayan culture<br>No commitment from the administration to hire teachers with specific training |
| Individuals' and society's attitude for recognizing Mayan culture | Internal processes | Dignify being Mayan<br>Being Mayan means respecting others |
| | External processes | The administration tends to convince the Mayans that society is not for them<br>No government support<br>Mayan representation in governments<br>Critical future for the Maya peoples<br>Lack of funding<br>Make the culture known to be recognized even in non-Mayan environments |

Source: authors, based on data analysis.

### 4.1. Mayan Culture, Cosmovision, and Language: Rejection and Hardship as Obstacles to the Construction of Personal Identity in Guatemala

4.1.1. Mayan Culture

When referring to the Mayan culture and cosmovision, it is important to start by defining these terms, as they can often be confusing for those who are not familiar with them. In the Guatemalan context, the significance of cosmovision and its influence on the culture and way of life of the population becomes evident, particularly in rural areas [25]. However, upon closer examination, we can observe that there are a few differences in the symbolic imagination of most participants. Being Mayan is not solely defined by belonging or birth; it encompasses a comprehensive essence, a feeling, a way of life, or, in other words, "the foundation of culture, so to speak" (i: Mayel).

> *"it means the food, the times, the way of thinking, of relating to the community, to other people . . . and then it also implies eh . . . as it has been a civilization based on agriculture, but not only. So farming times are important to understand oneself and nature".*

(i: Nohek)

However, when delving deeper, we observe that even though most of the interviewees emphasize that being Mayan is, above all, a feeling, it also involves the rituals "that make the culture something alive, it is something only a few know" (i: Luluknak), typical clothing, as one person stated, "my clothing, that is, my costume makes me a Maya woman, it identifies me" (i: Muun), and meals, as one person stated, "without a tortilla or a little bit of frijolito, we feel empty, even if we have eaten two, three, slices of bread, we feel that something is

missing" (i: Yunuen). Therefore, rituals, clothing, and meals can be understood as elements of identity that must be defended and reinforced within institutions and families.

4.1.2. Rejection

As several of the interviewees explained, although there is a particular current movement of vindication and recognition of the Mayan culture, which has been promoted since the Peace Accords (1991–1996), for many years in Guatemala, a significant percentage of the Indigenous population have tried to hide, conceal, put aside, or even abandon their cultural identity, history, culture, and cosmovision as if they were something to be ashamed of rather than proudly defended. One interviewee stated the following:

*"That is how I am registered on the Civil Registry. Something that for 40 years I have carried as a burden. Twenty to thirty years ago, it was very difficult, even for my family, my uncles, and my neighbors. My name was changed because they thought that telling me my name meant that I would be offended, that I would be rejected, and, of course, a child does not understand. I still have trauma today, with rejection just because of my hair".*

(i: Yunuen)

Despite attracting a great deal of attention, this feeling of shame or rejection, as explained by the participants, was primarily related to the idea that public institutions, including the government of the nation and the dominant classes, were building and transmitting the idea that the Indigenous population was:

*"second class in the world, historically pigeonholed at a lower level, condemned to extinction, annihilated between wars, diseases, exploitation, territorial invasions, or in the face of forced cultural assimilation".* [26]

(p. 45)

Thus, as the participants in this research study insist, there are several spaces in which it is possible to observe how Guatemalans themselves are ashamed of their origins and publicly reject them, mainly due to the perception that they are building of themselves, as stated in the following:

*"I looked around me, and there were indigenous people who no longer wanted to be indigenous, so they were already in crisis, first because they found the sense of being, of being . . . not indigenous and then because their parents were surely pressuring them to that . . . in the communities they say: study, mijo, so that you won't be like us. They understand the message, they don't want to be like their parents anymore, and when they graduate, they don't even invite their parents to come to their graduation";*

(i: Mayel)

This rejection is due to the language being used socially and causing boys and girls to move away from their roots gradually, as stated in the following:

*"I remember well that there was still talk about the Indian peoples [ . . . ], as a child we believed and were convinced that . . . it was not good to be an Indian, because to be an Indian was to be a fool. One was corrected by saying, "Don't be an Indian," which meant "Don't be a brute".*

(i: Yunuen)

Language is a controversial and challenging subject to deal with due to its complexity and variety but, at the same time, it occupies a very relevant differential category, as stated in the following:

*"Our brothers believe that by speaking Spanish, they will free themselves from their oppression, from their discrimination: "So they discriminate me less," "So they reject me less".*

(i: Mayel)

Although the issue of language will continue to appear throughout the text as it is a particularly relevant topic in the process of building the cultural identity of all peoples, it has been identified for decades as an inhibitor to social mobility. It has led many Mayan families to choose not to teach their children their mother tongue, the language of the Mayan people, with the understanding that if they knew Spanish first and used it as a vehicular language, it could become a space for transformation and social mobility.

Finally, it should be noted that, despite the rejection and long-term abandonment of everything that being Mayan implies, it should also be highlighted that some interviewees pointed out that recognizing the Mayan culture continues to be an urgent necessity even among those who, a priori, do not want to recognize it.

*" . . . people . . . see us as a source, not only for consultation, but also as a source of reference, right? I mean, one comes and is asked in the market what cosmovision is, and people will say, "I don't know," but if asked, " . . . do you burn your candle, do you thank . . . , ah . . . do you believe in . . . ;*

(i: Canek)

This either happens because the Mayan culture is synonymous with ignorance and inferiority or because it has a controversial relationship with the Christian religion, as follows:

*"So . . . in one of my last conversations with a girl, she tells me: "Teacher . . . I don't like what you are saying. You offend my religion, so I am going to tell my grandmother . . . Ah! A few days later, she returned and said, "Teacher, look, I told my grandmother what you told us about the Mayas and everything . . . and . . . do you know what my grandmother said to me? My grandmother began to pray for you. And (laughs), and then I just said thank you, right?".*

(i: Yunuen)

### 4.2. The School: Lights and Shadows of a Space for Identity Construction

#### 4.2.1. Mayan Languages

Returning to the question of language, this time from the school's perspective, it is clear from the interviews that although bilingual intercultural education has been a national priority in Guatemala for over a decade [27], Spanish continues to be the predominant language in schools. This not only hinders the school from becoming the privileged space for identity construction that it should be, but also alienates Mayan children at times. The standardized Spanish they encounter in school differs from the language that is spoken in their homes, posing obstacles not only to their identity formation, but also to their basic education.

This hindrance sometimes occurred because they did not understand what was required of them, causing the protagonists themselves to highlight how they spent their days in the schools with little knowledge of what was going on and how they developed strategies to survive, as stated in the following:

*"The children of the bosses or the foremen on the farm [ . . . ] they speak Spanish among themselves, don't they? We would get together in small groups to speak Q'eqchi' [ . . . ]. It cost us a lot, so we did not achieve an education, but rather a poor one . . . An anecdote that I remember well is that he sent me to bring a stapler [ . . . ] and . . . I half understood, [ . . . ] and on the way, I forgot the word [ . . . ]. And that is when . . . the teachers scolded us: "You are fools, you don't remember anything, go again".*

(i: Mayel)

In other instances, because they did not feel recognized, they were even forced to drop out of school, driven more by demotivation and the shame of not living up to a monocultural and exclusive system than by academic, attitudinal, or aptitudinal issues, as stated in the following:

*" . . . my mother tongue is Q'eqchi', but years ago there was no bilingual education, so they only taught the courses in Spanish, [ . . . ] it took me a long time to adapt and . . .*

*for the same reason I think I dropped out and when I came back I had to repeat a year. I didn't understand what the teachers were saying, no, I didn't understand things".*

(i: Anayansi)

Furthermore, the interviewees made it clear that schools, due to their structure and tradition, do not really recognize the Maya peoples, their culture, or cosmovision, not only excluding their uses, customs, and rituals, but also leaving out relevant contents in the curricula, and relegating the contents that are occasionally worked on to a folkloric position, reminding us of the tourist curriculum with which [28] used to define intercultural education in Spanish schools thirty years ago, when topics related to internal and external cultural differences were dealt with from the perspective of memory, exoticism, trivialization, and banalization. In this regard, the same author stated the following:

*"School institutions are places of struggle, and pedagogy can and must be a form of political-cultural struggle. The mission of educational centers as institutions of socialization is to expand human capacities, favor analysis and processes of common reflection of reality, and develop in students the procedures and skills essential for their responsible, critical, democratic and solidary action in society. All teachers need to be involved in the creation of alternative education models. One of the ways to start may be by constructing curricular materials capable of questioning current injustices and unequal social relations".* [28]

(p. 63)

### 4.2.2. A Teacher's Implication

Throughout the data collection process, and despite what has been stated so far, almost all the interviewees evoked a turning point in their Mayan identities in schools. On all occasions, this memory is related to the arrival/encounter/intervention of a teacher who helped them to recognize themselves, approaching in one way or another the cultural identity they "brought from home" without it seeming or becoming inhibiting and excluding.

Occasionally, a teacher helped them regain confidence in school as a safe space for expressing and constructing their personal identity.

*"Well, the truth is that when I decided to go back to school, I no longer had the same teacher, the new one was more aware of us, and I had other classmates who . . . no one bullied me, that is, the ones who had, were no longer in the same class. So maybe that is why I felt safer. Nobody discriminated against me; nobody excluded me from the group, everything was . . . different".*

(i: Anayansi)

In some instances, teachers became a reference point for rediscovering cultural traits that, even in the domestic sphere, people tried to hide and forget, as it was identified as a sign of a lack of culture, as one person stated,

*" . . . And so a teacher invited me in front of an audience when I was in high school to express myself, and, from that moment on, I understood why I had to rescue my self-esteem, right? And then with the family, because the family decided to . . . move towards other, towards other ideological and religious currents . . . until the teacher, when I was already in high school, at the age of 12, made me reflect and from then on my struggle began".*

(i: Yunuen)

In other instances, teachers were "saviors" and guides who not only allowed them to rediscover school as a place of personal development, but continued to be role models throughout their lives in their own professional careers, as one person stated,

*"It wasn't until 1972 that I started learning . . . I learnt to write through a Belgian priest here in Chamelco in an already diversified school. We began to talk about it, then the priest began to talk to us about conferences, and finally he invited me to work with him.*

*Then he was the one who was building me up and saying: you are worthy, you have your language, you have your culture, eh ... he was an expert in Mayan culture, so that is how the tables were turned, and when I became a teacher, I went to teach in some communities in Cahabón".*

(i: Mayel)

However, all the interviewees clarified how personal initiatives and individual actions have been responsible for their approach to and recognition of the Mayan culture within the framework of educational institutions that should be built as privileged spaces.

Likewise, the interviewees emphasized that beyond the institutional support they receive within their classrooms and as part of the teaching and learning process, they are responsible for making the Mayan culture known and respected, and for transmitting and dignifying its being and existence for everyone.

*"What I am doing is to make my own contribution from where I am standing, within my context, so that ... later, it cannot be said that we did nothing, right? So, no, I try to do what I can, and I will gladly give you my contacts so arrangements can be made for another time".*

(i: Imox)

In addition, they also recognize that the superstructure of educational institutions makes both dissemination and dignification very difficult, not only because of the lack of administrative support, although it formally complies by financing intercultural bilingual schools and financially rewarding teachers who choose to attend them, but also because of the lack of individual commitment from teachers who sometimes accept these assignments more for the associated economic benefits rather than for their preparation and suitability for the position. In fact, according to our interviewees,

*"Teachers trained in the system also do not want to teach a Q'eqchi' child properly, from the heart. They only "half" teach Q'eqchi' but continue in Spanish. Eh ... and those teachers also do not master the writing of their mother tongue. It does not dominate because they were never taught how it should be. Some teachers do, but others just muddle through and fill a position".*

(i: Imox)

The provided description highlights two important aspects. Firstly, it reveals a lack of individual commitment from teachers who end up working in rural Mayan areas, which hinders any process of recognition and dissemination of the Mayan culture. Secondly, it exposes the failure of the administrations to fulfill their obligations by not exercising any real control over the training or knowledge of those appointed to these positions or the supervision over what is taught in intercultural bilingual schools.

Finally, it should be noted that school life represents a significant cultural shock for many of those who were interviewed. However, it also represents a stimulus for vindication and a space in which the search for peers in the same situation has made it possible to build alternative spaces for affirmation and recognition beyond their families, developing new cultural spaces from these synergies and encounters. Indeed, sharing this space with peers is undoubtedly especially relevant at various stages of development when, as highlighted by several studies [29–31], the relationship with peers is more crucial for young people than the one they have established throughout their lives with their families. This is a space that, as several interviewees emphasized, is necessary, because otherwise, there would be a significant risk of the Mayan cultural identity being lost and diluted in the reality of the 21st century.

*4.3. Between the People and the Administration: Internal and External Processes for the Recognition of the Mayan Culture*

Delving further into our analysis, it is worth emphasizing that most of the participants, when speaking of cultural identity and the Maya peoples in the 21st century, were not only

referring to the actions of teachers, diverse sectors of the population, or the Maya peoples themselves in the defense of and commitment to their culture and cosmovision, as has been shown so far. One of the recurring elements in the analyzed data has to do with the idea of dignifying and honoring a culture that has been the reason for conflicts and discrimination for too long instead of being a source of pride and identification.

The participants consistently emphasized the need for the Mayan culture to not only be free from discrimination, but also to be preserved and protected in order to achieve dignity. This responsibility lies not only with the Maya peoples themselves, but also with the Guatemalan society and institutions. It is essential for society to move away from perceiving the Mayan culture, cosmovision, language, and traditional attire as representations of a marginalized and uneducated population. It is time for the Mayan culture to be recognized as a fundamental part of Guatemala's cultural heritage, which is rich in tradition, traditions, customs, and folklore.

> *"We seek the dignification of our people. Dignification, that we are dignified, that they look at us with respect, that they look at us with, with … as normal, common human beings and as it should be"* .

> (i: Mayel)

On the other hand, the interviewees emphasized the importance of respecting the "other" as a critical factor in the Mayan culture, including respect for the elders, the land, nature, and traditions. They also pointed out that it remains a factor of cultural identity for those who assume it, given that, in reality, to speak of the Mayan culture as if it were something antagonistic is only born of the lack of recognition and respect for it on the part of society.

> *"Personally, the essence of Maya culture is respect. It allows us to, uh … do several things. When we interact in a framework of respect, do business, interact with people in … sporting activities … everything works out better".*

> (i: Luluknak)

Both situations require the collaboration of public administrations, in general and educational administrations in particular, as well as the people who comprise them or the associations that promote them. However, if we pay attention to this, we quickly observe that even though there is some symbolic representation of Mayan citizens in governmental spaces, the institutional support for the recovery, dissemination, and dignification of the Mayan culture and identity, far from being led by decision makers, is organized by cultural or religious associations, and is almost always as a result of personal initiatives that end up leading working groups. There are three key elements to this debate as follows: (a) the genuine lack of institutional support that relegates cultural issues to a minority status; (b) the need to make the Mayan cosmovision known in order to obtain recognition, as well as the personal initiatives that result from it; and (c) the lack of means and resources, both economic and personal, to achieve this.

Key Elements to Clarify the Debate

Public administrations are little inclined to support policies that develop the Mayan culture, which is something that can be seen, among other things, in the fact that even though the Indigenous population is in the majority, only two or three of their representatives are in government, and are almost always in secondary positions designated fundamentally to silence possible claims of under-representation rather than with the real possibility of implementing policies of recognition, redistribution, and dignity [32].

> *" … since 1993 we began to see Mayas in the ministries … then the pure right-wing, those who shrunk the State, there were no Mayans in the state. And now, in the next government, there are Mayas again. But they all in the Ministry of Culture and Sports or something like that, we have been treated terribly".*

> (i: Canek)

These policies, if not managed in parallel with the institutions, may end up promoting economic inequality insofar as they, as Fraser [32] states, reify collective identities, running *"the risk of sanctioning the violation of human rights and freezing the antagonisms between those it seeks to mediate."* (p. 98)

Furthermore, according to some of the interviewees, Guatemalan administrations carry out policies that inhibit the defense of the cultural identities of the Maya peoples by not recognizing them as worthy and valuable. Public policies transmit to the population the idea of the need to overcome cultural identities as an element of social mobility, thus falling into the deconstruction of the concept of cultural identity, which, according to Hegel (1770–1831), requires the subjectivity of the person in relation to society, i.e., the subject ends up recognizing themself only in relation to "the *other*," with the recognition of others being essential for the development of a sense of self. Not being recognized or being recognized inadequately means *"suffering simultaneously a distortion in one's relationship with oneself and a damage inflicted against one's own identity"* [33] (p. 57).

At present, the Guatemalan national identity is more confused with the implementation and popularization of the term *"chapín"*, which is a type of footwear used by Creoles living in Santiago de Guatemala during colonial times. This term is now strongly promoted by the Creole and Ladino bourgeoisie as "the Guatemalan way of being" and, to a large extent, has become the demonym of Guatemala in an attempt to erase the different cultures and identities within the country [34,35].

Notwithstanding the above, and in line with what has been pointed out, there are groups of people in Guatemala who promote the Mayan culture as an element of cultural recognition, both inside and outside of schools. Thus, although schools are privileged places for this, it is necessary that in the absence of solid recognition policies, other social spheres also end up being showcases and places of debate, where those representatives of the Mayan culture, who we can somehow define as leaders, can make themselves known and recognized.

As reported in most of the interviews, a high percentage of people who are trying to promote the Mayan culture belong to cultural or religious associations operating in challenging environments. They explain that, in many cases, the rejection of the Mayan culture stems more from a lack of knowledge and dignification than from a lack of tradition, as follows:

> *"take the case of my abuelita . . . she knew about it . . . They would come and ask her for rituals, although later they would deny it, saying that it was heresy . . . but in the end, they all practice it . . . and when asked, they say that it is not that, but that they needed help".*

> (i: Imox)

Finally, it is important to emphasize that, as seen from the interviews, one of the greatest challenges for those trying to promote the Mayan culture or exercise political representation is the lack of funding. In Guatemala, it is essential to have sufficient financial means to, for example, run for political office.

> *"For example, in the Council, I think that . . . , say there are nine or ten seats available, . . . the first, second, third, but only two or three will make it into the Congress, so the last positions are the ones that are given to the indigenous people. When they have to contribute to financing their political party, then there is no money left to finance, uh . . . the price of the seats. It depends on which party it is; it could be one million, the second 800,000, the third 700,000 and so on. [ . . . ] So it will be almost impossible to participate".*

> (i: Mayel)

Similarly, in the school context, the interviewees stressed that it is not only necessary to be trained and to educate, but that funding is also required, as without it, the work of a few will end up being sporadic and vague.

*"When you propose something, it does not do much good, does it? We do not have the financial means to mobilise, it requires resources. Churches and other sectors have the resources, and they use them, right? So, the Don Bosco center, which is eminently Salesian, easily has 500 students. There they have power while we, compañeros de tortilla (literally, maize comrades, do not".*

(i: Nohek)

This lack of funding left the interviewees feeling defeated and challenged vis-à-vis the struggle of the Maya peoples to regain their rightful place in the Guatemalan society.

*"I have several cousins here in Chamelco, uh . . . they just don't want to get involved in anything, not in athletics, not in parties, they just go to the chapel, and that is where the young people are. So, everyone is singing and clapping, but in the end, what is behind all that and what is the future of these young people? From the Maya point of view, it is critical".*

(i: Luluknak)

## 5. Conclusions

To conclude this paper, we will highlight the key themes and ideas that have served to respond to the proposed objectives since, despite having been carried out in a concrete space of action, the use of the naturalistic interpretive research paradigm will allow for the reader to transfer many of the inferred findings [18] to other situations in which the focus is on issues of ethnicity, cultural identity, and school in Mayan contexts. Hence, our conclusions are presented as evidence to continue reflecting and deepening our understanding with new horizons rather than as closed and immovable axioms.

Firstly, it has become evident that the discussions and practices surrounding cosmovision, the Mayan culture, rituals, and Indigenous languages as elements of identity construction are increasingly important. For a long time, aspects related to ethnicity have been undervalued [36], hidden, and laden with negative symbolism, relegating the Maya peoples to a secondary status level. Thus, the participants, aware of this reality, have emphasized, time and again, the need to recover these spaces of identification, not only from the possibility of making themselves known, but, above all, from the idea of respect for the "others." It has been claimed that what is most important for the Maya peoples to return to (re)find themselves in their origins is the need for a space to (re)create identity. This (re)creation depends on what Fraser [32] defined as the need for citizen recognition and the dignification of the values found. Recognizing oneself and being recognized as part of the Maya peoples, both publicly and privately, without implying discrimination and exclusion, must be the objective of personal, social, and political demands. As other studies have shown [30,37–40], the construction of identity in an environment in which one participates that causes discrimination and social exclusion greatly complicates personal and social development.

Secondly, schools have transitioned from being hostile places for those expressing their ethnic identity to being potential spaces for inclusive and intercultural socialization. The problem is that despite the approval of the creation of intercultural bilingual schools with benefits for teachers who become involved in them, structural challenges, such as the lack of teacher training, the monolithic nature of the curriculum, and the overarching school framework, hinder schools from becoming privileged spaces for identity recognition. Thus, the reality continues to be that of institutions that promote the "White" and "Western" cosmovision, which is only relevant to a few and moves away from the ethnic culture. One way to rethink the shool's structure is to think of it in terms of "interculturalism" that, as several studies also indicate [41–45], does not allow for genuine change, but is what Santos Guerra [46] would define as a change that is more about the continent than the content.

Finally, when paying close attention to the explicit and implicit meanings of educational policies and how these influence the issues addressed, it should be emphasized here that two transversal factors mediate both how the Maya peoples construct their identity

and the influence that the school can have as a result of the educational policies that are implemented. Thus, it has become evident that, while it is true that structural changes should be global and not particularistic, currently, awareness-raising and dissemination initiatives depend more on the personal initiatives of leaders in the framework of nongovernmental organizations or education centers, including schools, in which they voluntarily and willingly assume this role, than on the operation of plans and projects. The lack of involvement of the majority of those who work in multicultural environments inhibits the impact of social and educational policies, with the responsibility falling on teachers and principals who, in most cases, have neither the training nor the financial means to implement these policies. On the other hand, it is necessary to point out that the lack of control and supervision related to the implementation of the approved norms concerning the dissemination of the culture and protection of Mayan languages shows how educational administrations legislate more to comply with acquired commitments than for the actual recognition of a people and their culture, setting in motion dynamics that not only lose their potential, but are blurred between established power strategies and insufficient funding.

**Author Contributions:** Conceptualization, S.F.-L., M.R. and J.G.-M.; methodology, S.F.-L.; validation, S.F.-L., M.R. and J.G.-M.; formal analysis, M.R. and J.G.-M.; investigation, GI-HUM413; resources, J.G.-M.; data curation, J.G.-M.; writing—original draft preparation, M.R., S.F.-L. and J.G.-M.; writing—review and editing, M.R.; visualization, J.G.-M.; supervision, S.F.-L.; project administration, S.F.-L.; funding acquisition, S.F.-L. All authors have read and agreed to the published version of the manuscript.

**Funding:** This research was funded by Agencia Andaluza de Cooperación Internacional para el Desarrollo (AAECID). International Cooperation Project cod. 2018UC006.

**Institutional Review Board Statement:** Ethical review and approval was not required for the study on human participants in accordance with the local legislation and institutional requirements. However, we have maintained ethics with constant negotiation with the participants.

**Informed Consent Statement:** In line with the research paradigm, we negotiate with all the participants the access, process and data obtained, as well as triangulation processes.

**Data Availability Statement:** Contact the researchers.

**Conflicts of Interest:** The authors declare no conflict of interest.

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
