# Peer review of "To Make Known in Order to Recognize: Schools as Vehicles for Constructing Identity for the Maya Peoples"

_education, doi:10.3390/educsci13070743_

Round 1
Reviewer 1 Report
This is a very interesting and well-written paper exploring the role of schools in constructing identity for the Maya peoples. The research is extremely important because, as far as I know, the research on this specific issue is extremely limited.Both the theoretical and the methodological part of the paper have a very good structure and there are absolutely clear of what they describe. I found the literature review very adequate as it contains many significant references focusing on the main subject of the paper, namely the construction of Maya identity and the role of education on that. Also, the methodology is described in detail giving a concrete picture of the design and implementation of the steps that the researchers followed. The study employed a qualitative design to collect data, which is the most appropriate for this type of research.
The description of the research results is clear and the discussion/conclusion is extremely fruitful. I found very interesting the point that “what is most important for the Maya people to return to (re)find themselves in their origins is the need for a space to (re)create identity”, a point which is food for thought for the new role which the schools have to play.
As a conclusion I believe that this an excellent paper giving many ideas and motives to future researchers on the same and similar fields.
Author Response
Thank you very much for the feedback. We have been working with the Maya indigenous communities for many years, and we believe that it is a pressing issue that can be generalized to other educational settings due to its significant impact.
Reviewer 2 Report
The subject is very compelling but the exposition is not clear. There are assumptions unstated and the organization is difficult to follow. The use of quotes is important but there linkages and sequence is overwhelming. The ideas need to be outlined clearly and organized in a way that the points they wish to bring out are clearly stated and supported.
I am not sure about the command of English but am more confident that the exposition lacks organization and even with problems in English a good outline may be able to address these issues. As it is the themes are important but get lost in the way it is presented.
Author Response
Dear reviewer, thank you for your suggestions. First of all, we understand that since it was originally written in Spanish, its structure may have made translation and clarity of writing difficult, so we have reviewed the whole text with the translator and tried to improve it.
In the same way, and always with the idea of improving the organisation and making it clearer, we have reorganised section 4 of the discussion of the results, adding a table in which the categories of analysis are organised and the subcategories are made explicit; as well as dividing points 4.1, 4.2 and 4.3 into many other subsections.
Finally, we have included some references and theoretical explanations in the first part, reviewed the methodology and checked the references cited.
We hope to have improved the online writing with your suggestions.
Yours sincerely
Reviewer 3 Report
In the chapter 2 as a theoretical background, it is necessary to add theoretical content on the relation between school (teacher) and identity formation.
When presenting research results in the chapter 4, it is recommended to summarize them with one table, which is categorized with major and minor categories. Section 4.1, 4.2, and 4.3 are long and consisting of two opposing terms respectively, so it is better to subdivide them. For example, 4.1 can be divided into two categories: 4.1.1 Mayan culture, 4.1.2 Rejection
After finishing the text, the source of the research funds for this study should be separately revealed.
Sentences 556-557, and 526 are not proper.
Author Response
Dear revisor, thank you very much for the suggestion. In order to that we have:
-. included theoretical content on the relation between school (teacher) and identity formation as a theoretical background in section 2.
-. Created a table to summarize result divided into mayor and minor categories.
-. Subdivided section 4 in:
4.1. Mayan culture, cosmovision and language. Rejection and hardship as obstacles to the construction of personal identity in Guatemala
4.1.1. Mayan Culture
4.1.2 Rejection
4.2. The school. Lights and Shadows of a space for identity construction
4.2.1 Mayan languages
4.2.2 Teachers implication
4.3. Between the personal and the administration. Internal and external processes for the recognition of Mayan culture
4.3.1 Key elements to clarify the debate
-. ReviewedSentences 556-557
-. Reviewed all cited references.
Hoping to have solved and improved the paper, I greet you cordially.